# Semi-parametric Topological Memory for Navigation

**Nikolay Savinov**[*]
ETH Zürich

**Alexey Dosovitskiy**[*]
Intel Labs

**Vladlen Koltun**
Intel Labs

## Abstract

We introduce a new memory architecture for navigation in previously unseen environments, inspired by landmark-based navigation in animals. The proposed semi-parametric topological memory (SPTM) consists of a (non-parametric) graph with nodes corresponding to locations in the environment and a (parametric) deep network capable of retrieving nodes from the graph based on observations. The graph stores no metric information, only connectivity of locations corresponding to the nodes. We use SPTM as a planning module in a navigation system. Given only 5 minutes of footage of a previously unseen maze, an SPTM-based navigation agent can build a topological map of the environment and use it to confidently navigate towards goals. The average success rate of the SPTM agent in goal-directed navigation across test environments is higher than the best-performing baseline by a factor of three.

## 1    Introduction

Deep learning (DL) has recently been used as an efficient approach to learning navigation in complex three-dimensional environments. DL-based approaches to navigation can be broadly divided into three classes: purely reactive (Dosovitskiy & Koltun, 2017; Zhu et al., 2017), based on unstructured general-purpose memory such as LSTM (Mnih et al., 2016; Mirowski et al., 2017), and employing a navigation-specific memory structure based on a metric map (Parisotto & Salakhutdinov, 2018; Gupta et al., 2017).

However, extensive evidence from psychology suggests that when traversing environments, animals do not rely strongly on metric representations (Gillner & Mallot, 1998; Wang & Spelke, 2002; Foo et al., 2005). Rather, animals employ a range of specialized navigation strategies of increasing complexity. According to Foo et al. (2005), one such strategy is landmark navigation – "the ability to orient with respect to a known object". Another is route-based navigation that "involves remembering specific sequences of positions". Finally, map-based navigation assumes a "survey knowledge of the environmental layout", but the map need not be metric and in fact it is typically not: "[...] humans do not integrate experience on specific routes into a metric cognitive map for navigation [...] Rather, they primarily depend on a landmark-based navigation strategy, which can be supported by qualitative topological knowledge of the environment."

In this paper, we propose semi-parametric topological memory (SPTM) – a deep-learning-based memory architecture for navigation, inspired by landmark-based navigation in animals. SPTM consists of two components: a non-parametric memory graph $G$ where each node corresponds to a location in the environment, and a parametric deep network $R$ capable of retrieving nodes from the graph based on observations. The graph contains no metric relations between the nodes, only connectivity information. While exploring the environment, the agent builds the graph by appending observations to it and adding shortcut connections based on detected visual similarities. The network $R$ is trained to retrieve nodes from the graph based on an observation of the environment. This allows the agent to localize itself in the graph. Finally, we build a complete SPTM-based navigation agent by complementing the memory with a locomotion network $L$, which allows the agent to move between nodes in the graph. The $R$ and $L$ networks are trained in self-supervised fashion, without any manual labeling or reward signal.

---

[*]Shared first authorship

We evaluate the proposed system and relevant baselines on the task of goal-directed maze navigation in simulated three-dimensional environments. The agent is instantiated in a previously unseen maze and given a recording of a walk through the maze (images only, no information about actions taken or ego-motion). Then the agent is initialized at a new location in the maze and has to reach a goal location in the maze, given an image of that goal. To be successful at this task, the agent must represent the maze based on the footage it has seen, and effectively utilize this representation for navigation.

The proposed system outperforms baseline approaches by a large margin. Given 5 minutes of maze walkthrough footage, the system is able to build an internal representation of the environment and use it to confidently navigate to various goals within the maze. The average success rate of the SPTM agent in goal-directed navigation across test environments is higher than the best-performing baseline by a factor of three. Qualitative results and an implementation of the method are available at `https://sites.google.com/view/SPTM`.

## 2    RELATED WORK

Navigation in animals has been extensively studied in psychology. Tolman (1948) introduced the concept of a cognitive map – an internal representation of the environment that supports navigation. The existence of cognitive maps and their exact form in animals, including humans, has been debated since. O'Keefe & Nadel (1978) suggested that internal representations take the form of metric maps. More recently, it has been shown that bees (Cartwright & Collett, 1982; Collett, 1996), ants (Judd & Collett, 1998), and rats (Sutherland et al., 1987) rely largely on landmark-based mechanisms for navigation. Bennett (1996) and Mackintosh (2002) question the existence of cognitive maps in animals. Gillner & Mallot (1998), Wang & Spelke (2002), and Foo et al. (2005) argue that humans rely largely on landmark-based navigation.

In contrast, navigation systems developed in robotics are typically based on metric maps, constructed using the available sensory information – sonar, LIDAR, RGB-D, or RGB input (Elfes, 1987; Thrun et al., 2005; Durrant-Whyte & Bailey, 2006). Particularly relevant to our work are vision-based simultaneous localization and mapping (SLAM) methods (Cadena et al., 2016). These systems provide high-quality maps under favorable conditions, but they are sensitive to calibration issues, do not deal well with poor imaging conditions, do not naturally accommodate dynamic environments, and can be difficult to scale.

Modern deep learning (DL) methods allow for end-to-end learning of sensorimotor control, directly predicting control signal from high-dimensional sensory observations such as images (Mnih et al., 2015). DL approaches to navigation vary both in the learning method – reinforcement learning or imitation learning – and in the memory representation. Purely reactive methods (Dosovitskiy & Koltun, 2017; Zhu et al., 2017) lack explicit memory and do not navigate well in complex environments (Savva et al., 2017). Systems equipped with general-purpose LSTM memory (Mnih et al., 2016; Pathak et al., 2017; Jaderberg et al., 2017; Mirowski et al., 2017) or episodic memory (Blundell et al., 2016; Pritzel et al., 2017) can potentially store information about the environment. However, these systems have not been demonstrated to perform efficient goal-directed navigation in previously unseen environments, and empirical results indicate that LSTM-based systems are not up to the task (Savva et al., 2017). Oh et al. (2016) use an addressable memory for first-person-view navigation in three-dimensional environments. The authors demonstrate that the proposed memory structure supports generalization to previously unseen environments. Our work is different in that Oh et al. (2016) experiment with relatively small discrete gridworld-like environments, while our approach naturally applies to large continuous state spaces.

Most related to our work are DL navigation systems that use specialized map-like representations. Bhatti et al. (2016) augment a DL system with a metric map produced by a standard SLAM algorithm. Parisotto & Salakhutdinov (2018) use a 2D spatial memory that represents a global map of the environment. Gupta et al. (2017) build a 2D multi-scale metric map using the end-to-end trainable planning approach of Tamar et al. (2016). Our method differs from these approaches in that we are not aiming to build a global metric map of the environment. Rather, we use a topological map. This allows our method to support navigation in a continuous space without externally provided camera poses or ego-motion information.

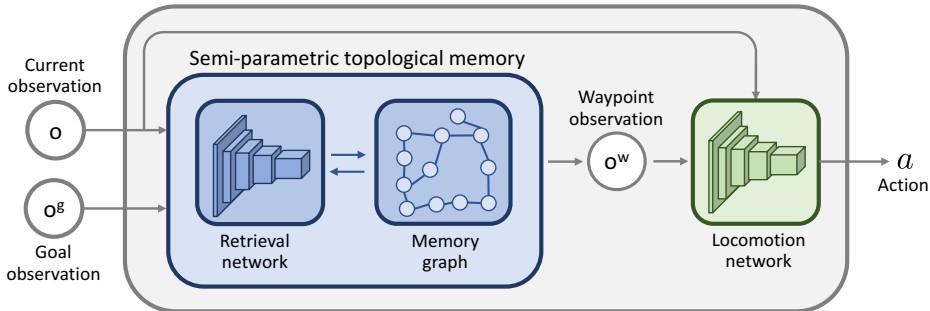

Figure 1: A navigation agent equipped with semi-parametric topological memory (SPTM). Given the inputs – the current observation $\mathbf{o}$ and the goal observation $\mathbf{o}^g$ – SPTM provides a waypoint observation $\mathbf{o}^w$. This waypoint and the current observation $\mathbf{o}$ are fed into the locomotion network $L$, which outputs the action $a$ to be executed in the environment.

While contemporary approaches in robotics are dominated by metric maps, research on topological maps has a long history in robotics. Models based on topological maps have been applied to navigation in simple 2D mazes (Kuipers & Byun, 1991; Meng & Kak, 1993; Schölkopf & Mallot, 1995) and on physical systems (Hong et al., 1992; Bachelder & Waxman, 1995; Franz et al., 1998; Thrun, 1998; Fraundorfer et al., 2007). Trullier et al. (1997) provide a review of biologically-inspired navigation systems, including landmark-based ones. Milford and colleagues designed SLAM systems inspired by computational models of the hippocampus (Milford & Wyeth, 2008; 2010; Ball et al., 2013). We reinterpret this line of work in the context of deep learning.

## 3 METHOD

We consider an agent interacting with an environment in discrete time steps. At each time step $t$, the agent gets an observation $\mathbf{o}_t$ of the environment and then takes an action $a_t$ from a set of actions $\mathcal{A}$. In our experiments, the environment is a maze in a three-dimensional simulated world, and the observation is provided to the agent as a tuple of several recent images from the agent's point of view.

The interaction of the agent with a new environment is set up in two stages: exploration and goal-directed navigation. During the first stage, the agent is presented with a recording of a traversal of the environment over a number of time steps $T_e$, and builds an internal representation of the environment based on this recording. In the second stage, the agent uses this internal representation to reach goal locations in the environment. This goal-directed navigation is performed in an episodic setup, with each episode lasting for a fixed maximum number of time steps or until the goal is reached. In each episode, the goal location is provided to the agent by an observation of this location $\mathbf{o}^g$. The agent has to use the goal observation and the internal representation built during the exploration phase to effectively reach the goal.

### 3.1 SEMI-PARAMETRIC TOPOLOGICAL MEMORY

We propose a new form of memory suitable for storing internal representations of environments. We refer to it as semi-parametric topological memory (SPTM). It consists of a (non-parametric) memory graph $G$ where each node represents a location in the environment, and a (parametric) deep network $R$ capable of retrieving nodes from the graph based on observations. A high-level overview of an SPTM-based navigation system is shown in Figure 1. Here SPTM acts as a planning module: given the current observation $\mathbf{o}$ and the goal observation $\mathbf{o}^g$, it generates a waypoint observation $\mathbf{o}^w$, which lies on a path to the goal and can be easily reached from the agent's current location. The current observation and the waypoint observation are provided to a locomotion network $L$, which is responsible for short-range navigation. The locomotion network then guides the agent towards the waypoint, and the loop repeats. The networks $R$ and $L$ are trained in self-supervised fashion,

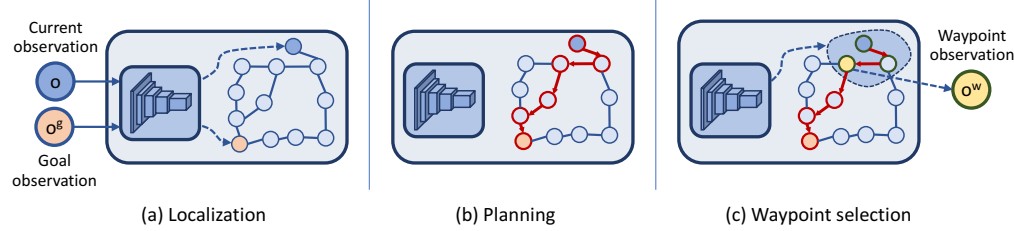

Figure 2: The use of semi-parametric topological memory for navigation. (a) The retrieval network $R$ localizes in the graph the vertices $v^a$ (blue) and $v^g$ (orange), corresponding to the current agent's observation $\mathbf{o}$ and the goal observation $\mathbf{o}^g$, respectively. (b) The shortest path on the graph between these vertices is computed (red arrows). (c) The waypoint vertex $v^w$ (yellow) is selected as the vertex in the shortest path that is furthest from the agent's vertex $v^a$ but can still be confidently reached by the agent. The output of the SPTM is the corresponding waypoint observation $\mathbf{o}^w = \mathbf{o}_{v^w}$.

without any externally provided labels or reinforcement signals. We now describe each component of the system in detail.

**Retrieval network.** The network $R$ estimates the similarity of two observations $(\mathbf{o}_1, \mathbf{o}_2)$. The network is trained on a set of environments in self-supervised manner, based on trajectories of a randomly acting agent. Conceptually, the network is trained to assign high similarity to pairs of observations that are temporally close, and low similarity to pairs that are temporally distant. We cast this as a classification task: given a pair of observations, the network has to predict whether they are temporally close or not.

To generate the training data, we first let a random agent explore the environment, resulting in a sequence of observations $\{\mathbf{o}_1, \dots \mathbf{o}_N\}$ and actions $\{a_1, \dots a_N\}$. We then automatically generate training samples from these trajectories. Each training sample is a triple $\langle \mathbf{o}_i, \mathbf{o}_j, y_{ij} \rangle$ that consists of two observations and a binary label. Two observations are considered close ($y_{ij} = 1$) if they are separated by at most $l = 20$ time steps: $|i - j| \leq l$. Negative examples are pairs where the two observations are separated by at least $M \cdot l$ steps, where $M = 5$ is a constant factor that determines the margin between positive and negative examples.

We use a siamese architecture for the network $R$, akin to Zagoruyko & Komodakis (2015). Each of the two input observations is first processed by a deep convolutional encoder based on ResNet-18 (He et al., 2016), which outputs a 512-dimensional embedding vector. These two vectors are concatenated and further processed by a small 5-layer fully-connected network, ending with a 2-way softmax. The network is trained in supervised fashion with the cross-entropy loss. Further details are provided in the supplement.

**Memory graph.** The graph is populated based on an exploration sequence provided to the agent. Denote the observations in the sequence by $(\mathbf{o}_1^e, \dots, \mathbf{o}_{T_e}^e)$. Each vertex $v_i$ in the graph stores an observation of the environment, $\mathbf{o}_{v_i} = \mathbf{o}_i^m$. Two vertices $v_i$ and $v_j$ are connected by an edge in one of two cases: if they correspond to consecutive time steps, or if the corresponding observations are very close, as judged by the retrieval network $R$:

$$e_{ij} = 1 \iff |i - j| = 1 \ \lor \ R(\mathbf{o}_{v_i}, \mathbf{o}_{v_j}) > s_{shortcut}, \tag{1}$$

where $0 < s_{shortcut} < 1$ is a similarity threshold for creating a shortcut connection. The first type of edge corresponds to natural spatial adjacency between locations, while the second type can be seen as a form of loop closure.

Two enhancements improve the quality of the graph. First, we only connect vertices by a "visual shortcut" edge if $|i-j| > \Delta T_\ell$, so as to avoid adding trivial edges. Second, to improve the robustness of visual shortcuts, we find these by matching sequences of observations, not single observations:

$$\mathrm{median}\{R(\mathbf{o}_{v_{i-\Delta T_w}}, \mathbf{o}_{v_{j-\Delta T_w}}), \dots, R(\mathbf{o}_{v_{i+\Delta T_w}}, \mathbf{o}_{v_{j+\Delta T_w}})\} > s_{shortcut}. \tag{2}$$

**Finding the waypoint.** At navigation time, we use SPTM to provide waypoints to the locomotion network. As illustrated in Figure 2, the process includes three steps: localization, planning, and waypoint selection.

In the localization step, the agent localizes itself and the goal in the graph based on its current observation $\mathbf{o}$ and the goal observation $\mathbf{o}^g$, as illustrated in Figure 2(a). We have experimented with two approaches to localization. In the basic variant, the agent's location is retrieved as the median of $k = 5$ nearest neighbors of the observation in the memory. The siamese architecture of the retrieval network allows for efficient nearest neighbor queries by pre-computing the embeddings of observations in the memory.

An issue with this simple technique is that localization is performed per frame, and therefore the result can be noisy and susceptible to perceptual aliasing – inability to discriminate two locations with similar appearance. We therefore implement a modified approach allowing for temporally consistent self-localization, inspired by localization approaches from robotics (Milford & Wyeth, 2012). We initially perform the nearest neighbor search only in a local neighborhood of the previous agent's localization, and resort to global search in the whole memory only if this initial search fails (that is, the similarity of the retrieved nearest neighbor to the current observation is below a certain threshold $s_{local}$). This simple modification improves the performance of the method while also reducing the search time.

In the planning step, we find the shortest path on the graph between the two retrieved nodes $v^a$ and $v^g$, as shown in Figure 2 (b). We used Dijkstra's algorithm in our experiments.

Finally, the third step is to select a waypoint on the computed shortest path, as depicted in Figure 2(c). We denote the shortest path by

$$\langle v_0^{sp}, v_1^{sp}, \ldots, v_n^{sp} \rangle, \; v_0^{sp} = v^a, \; v_n^{sp} = v^g \tag{3}$$

A naive solution would be to set the waypoint to $v_D^{sp}$, with a fixed $D$. However, depending on the actions taken in the exploration sequence, this can lead to selecting a waypoint that is either too close (no progress) or too far (not reachable). We therefore follow a more robust adaptive strategy. We choose the furthest vertex along the shortest path that is still confidently reachable:

$$v^w = v_l^{sp}, \quad l = \max_i \{i, \text{ s.t. } R(\mathbf{o}, \mathbf{o}_{v_i^{sp}}) > s_{reach}\}, \tag{4}$$

where $0 < s_{reach} < 1$ is a fixed similarity threshold for considering a vertex reachable. In practice, we limit the waypoint search to a fixed window $i \in [H_{\min}, H_{\max}]$. The output of the planning process is the observation $\mathbf{o}^w = \mathbf{o}_{v^w}$ that corresponds to the retrieved waypoint.

## 3.2 LOCOMOTION NETWORK

The network $L$ is trained to navigate towards target observations in the vicinity of the agent. The network maps a pair $(\mathbf{o}_1, \mathbf{o}_2)$, which consists of a current observation and a goal observation, into action probabilities: $L(\mathbf{o}_1, \mathbf{o}_2) = \mathbf{p} \in \mathbb{R}^{|\mathcal{A}|}$. The action can then be produced either deterministically by choosing the most probable action, or stochastically by sampling from the distribution. In what follows we use the stochastic policy.

Akin to the retrieval network $R$, the network $L$ is trained in self-supervised manner, based on trajectories of a randomly acting agent. Random exploration produces a sequence of observations $\{\mathbf{o}_1, \ldots \mathbf{o}_N\}$ and actions $\{a_1, \ldots a_N\}$. We generate training samples from these trajectories by taking a pair of observations separated by at most $l = 20$ time steps and the action corresponding to the first observation: $((\mathbf{o}_i, \mathbf{o}_j), a_i)$. The network is trained in supervised fashion on this data, with a softmax output layer and the cross-entropy loss. The architecture of the network is the same as the retrieval network.

Why is it possible to learn a useful controller based on trajectories of a randomly acting agent? The proposed training procedure leads to learning the conditional action distribution $P(a|\mathbf{o}_t, \mathbf{o}_{t+k})$. Even though the trajectories are generated by a random actor, this distribution is generally not uniform. For instance, if $k = 1$, the network would learn actions to be taken to perform one-step transitions between neighboring states. For $k > 1$, training data is more noisy, but there is still useful training signal, which turns out to be sufficient for short-range navigation.

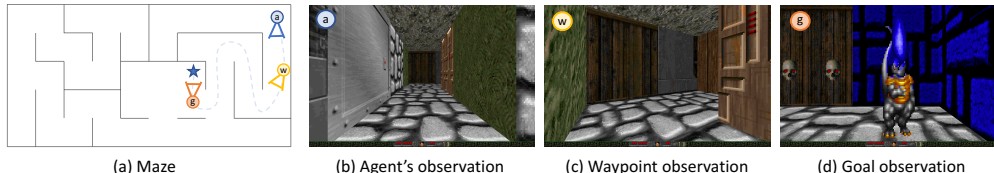

| (a) Maze | (b) Agent's observation | (c) Waypoint observation | (d) Goal observation |

Figure 3: SPTM-based agent navigating towards a goal in a three-dimensional maze (a). The agent aims to reach the goal, denoted by a star. Given the current agent's observation (b) and the goal observation (d), SPTM produces a waypoint observation (c). The locomotion network is then used to navigate towards the waypoint.

### 3.3 IMPLEMENTATION DETAILS

Inputs to the retrieval network $R$ and the locomotion network $L$ are observations of the environment $o$, represented by stacks of two consecutive RGB images obtained from the environment, at resolution $160 \times 120$ pixels. Both networks are based on ResNet-18 (He et al., 2016). Note that ResNet-18 is much larger than networks typically used in navigation agents based on reinforcement learning. The use of this high-capacity architecture is made possible by the self-supervised training of our model. Training of a network of this size from scratch with pure reinforcement learning would be problematic and, to our knowledge, has never been demonstrated.

The training setup is similar for both networks. We generate training data online by executing a random agent in the training environment, and maintain a replay buffer $B$ of recent samples. At each training iteration, we sample a mini-batch of $64$ observation pairs at random from the buffer, according to the conditions described in Sections 3.1 and 3.2. We then perform an update using the Adam optimizer (Kingma & Ba, 2015), with learning rate $\lambda = 0.0001$. We train the networks $R$ and $L$ for a total of $1$ and $2$ million mini-batch iterations, respectively. Further details are provided in the supplement.

We made sure that all operations in the SPTM are implemented efficiently. Goal localization is only performed once in the beginning of a navigation episode. Shortest paths to the goal from all vertices of the graph can therefore also be computed once in the beginning of navigation. The only remaining computationally expensive operations are nearest-neighbor queries for agent self-localization in the graph. However, thanks to the siamese architecture of the retrieval network, we can precompute the embedding vectors of observations in the memory and need only evaluate the small fully-connected network during navigation.

## 4 EXPERIMENTS

We perform experiments using a simulated three-dimensional environment based on the classic game Doom (Kempka et al., 2016). An illustration of an SPTM agent navigating towards a goal in a maze is shown in Figure 3. We evaluate the proposed method on the task of goal-directed navigation in previously unseen environments and compare it to relevant baselines from the literature.

### 4.1 SETUP

We are interested in agents that are able to generalize to new environments. Therefore, we used different mazes for training, validation, and testing. We used the same set of textures for all labyrinths, but the maze layouts are very different, and the texture placement is randomized. During training, we used a single labyrinth layout, but created $400$ versions with randomized goal placements and textures. In addition, we created $3$ mazes for validation and $7$ mazes for testing. Layouts of the training and test labyrinths are shown in Figure 4; the validation mazes are shown in the supplement. Each maze is equipped with $4$ goal locations, marked by $4$ different special objects. The appearance of these special objects is common to all mazes. We used the validation mazes for tuning the parameters of all approaches, and used fixed parameters when evaluating in the test mazes.

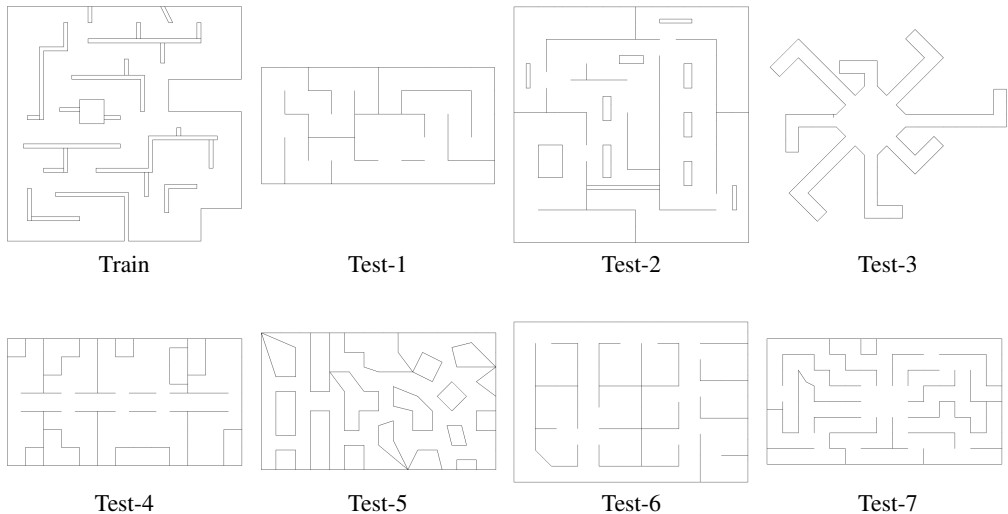

Figure 4: Layouts of training and test mazes.

The overall experimental setup follows Section 3. When given a new maze, the agent is provided with an exploration sequence of the environment, with a duration of approximately 5 minutes of in-simulation time (equivalent to 10,500 simulation steps). In our experiments, we used sequences generated by a human subject aimlessly exploring the mazes. The same exploration sequences were provided to all algorithms – the proposed method and the baselines. Example exploration sequences are shown on the project page, https://sites.google.com/view/SPTM.

Given an exploration sequence, the agent attempts a series of goal-directed navigation trials. In each of these, the agent is positioned at a new location in the maze and is presented with an image of the goal location. In our experiments, we used 4 different starting locations, 4 goals per maze, and repeated each trial 6 times (results in these vary due to the use of randomized policies for all methods), resulting in 96 trials for each maze. A trial is considered successfully completed if the agent reaches the goal within 5,000 simulation steps, or 2.4 minutes of in-simulation time.

### 4.1.1 HYPERPARAMETERS

We set the hyperparameters of the SPTM agent based on an evaluation on the validation set, as reported in Table S1 in the supplement. We find that the method performs well for a range of hyperparameter values. Interestingly, the approach is robust to temporal subsampling of the walkthrough sequence. Therefore, in the following experiments we subsample the walkthrough sequence by a factor of 4 when building the SPTM graph. Another important parameter is the threshold $s_{shortcut}$ for creating shortcuts in the graph. We set this threshold as a percentile of the set of all pairwise distances between observations in the memory, or, in other words, as the desired number of shortcuts to be created. We set this number to 2000 in what follows. When making visual shortcuts in the graph, we set the minimum shortcut distance $\Delta T_\ell = 5$ and the smoothing window size $\Delta T_w = 10$. The threshold values for waypoint selection are set to $s_{local} = 0.7$ and $s_{reach} = 0.95$. The minimum and maximum waypoint distances are set to $H_{\min} = 1$ and $H_{\max} = 7$, respectively.

### 4.2 BASELINES

We compare the proposed method to a set of baselines that are representative of the state of the art in deep-learning-based navigation. Note that we study an agent operating in a realistic setting: a continuous state space with no access to ground-truth information such as depth maps or ego-motion. This setup excludes several existing works from our comparison: the full model of Mirowski et al. (2017) that uses ground-truth depth maps and ego-motion, the method of Gupta et al. (2017) that operates on a discrete grid given ground-truth ego-motion, and the approach of Parisotto & Salakhutdinov (2018) that requires the knowledge of ground-truth global coordinates of the agent.

|                          | Test-1 | Test-2 | Test-3 | Test-4 | Test-5 | Test-6 | Test-7 | *mean* |
|--------------------------|--------|--------|--------|--------|--------|--------|--------|--------|
| Goal-agnostic feedforward | 26     | 25     | 27     | 23     | 32     | 20     | 20     | 24.7   |
| Goal-agnostic LSTM       | 53     | 39     | 45     | 18     | 27     | 36     | 19     | 33.9   |
| Goal-seeking feedforward | 30     | 24     | 29     | 18     | 24     | 27     | 22     | 24.9   |
| Goal-seeking LSTM        | 34     | 27     | 15     | 25     | 16     | 33     | 4      | 22     |
| **Ours**                 | **100** | **100** | **100** | **100** | **100** | **100** | **100** | **100** |

Table 1: Comparison of the SPTM agent to baseline approaches. We report the percentage of navigation trials successfully completed in 5,000 steps (higher is better).

The first baseline is a goal-agnostic agent without memory. The agent is not informed about the goal, but may reach it by chance. We train this network in the training maze using asynchronous advantage actor-critic (A3C) (Mnih et al., 2016). The agent is trained on the surrogate task of collecting invisible beacons around the labyrinth. (The beacons are made invisible to avoid providing additional visual guidance to the agents.) In the beginning of each episode, the labyrinth is populated with 1000 of these invisible beacons, at random locations. The agent receives a reward of 1 for collecting a beacon and 0 otherwise. Each episode lasts for 5,000 simulation steps. We train the agent with the A3C algorithm and use an architecture similar to Mnih et al. (2016). Further details are provided in the supplement.

The second baseline is a feedforward network trained on goal-directed navigation, similar to Zhu et al. (2017). The network gets its current observation, as well as an image of the goal, as input. It gets the same reward as the goal-agnostic agent for collecting invisible beacons, but in addition it gets a large reward of 800 for reaching the goal. This network can go towards the goal if the goal is within its field of view, but it lacks memory, so it is fundamentally unable to make use of the exploration phase. The network architecture is the same as in the first baseline, but the input is the concatenation of the 4 most recent frames and the goal image.

The third and fourth baseline approaches are again goal-agnostic and goal-directed agents, but equipped with LSTM memory. The goal-directed LSTM agent is similar to Mirowski et al. (2017). At test time, we feed the exploration sequence to the LSTM agent and then let it perform goal-directed navigation without resetting the LSTM state. When training these networks, we follow a similar protocol. First, the agent navigates the environment for 10,000 steps in exploration mode; that is, with rewards for collecting invisible beacons, but without a goal image given and with no reward for reaching a goal. Next, the agent is given a goal image and spends another 5,000 steps in goal-directed navigation mode; that is, with a goal image given and with a high reward for reaching the goal (while also continuing to receive rewards for collecting the invisible beacons). We do not reset the state of the memory cells between the two stages. This way, the agent can learn to store the layout of the environment in its memory and use it for efficient navigation.

## 4.3 RESULTS

Table 1 shows, for each test maze, the percentage of navigation trials successfully completed within 5,000 steps, equivalent to 2.4 minutes of real-time simulation. Figure 5 presents the results on the test mazes in more detail, by plotting the percentage of completed episodes as a function of the trial duration. Qualitative results are available at `https://sites.google.com/view/SPTM`.

The proposed SPTM agent is superior to the baselines in all mazes. As Table 1 demonstrates, its average success rate across the test mazes is three times higher than the best-performing baseline. Figure 5 demonstrates that the proposed approach is not only successful overall, but that the agent typically reaches the goal much faster than the baselines.

The difference in performance between feedforward and LSTM baseline variants is generally small and inconsistent across mazes. This suggests that standard LSTM memory is not sufficient to efficiently make use of the provided walkthrough footage. One reason can be that recurrent networks, including LSTMs, struggle with storing long sequences (Goodfellow et al., 2016). The duration of the walkthrough footage, 10,000 time steps, is beyond the capabilities of standard recurrent networks. SPTM is at an advantage, since it stores all the provided information by design.

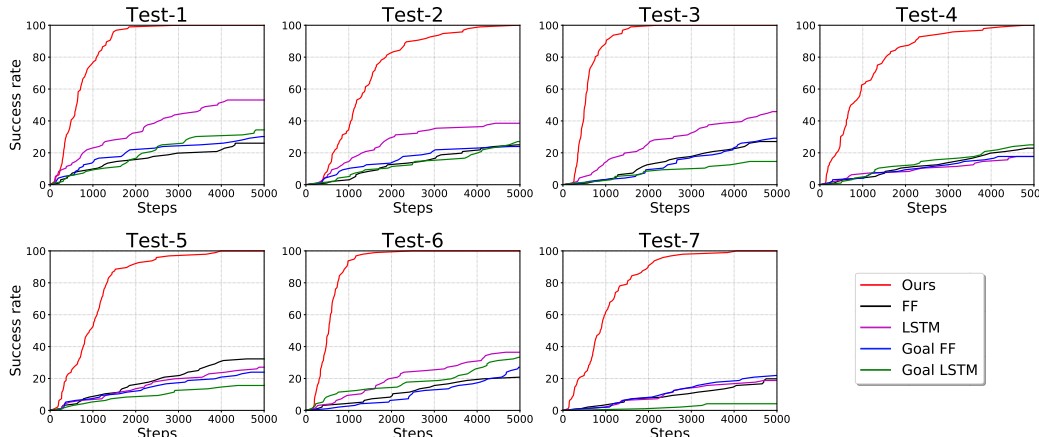

Figure 5: Percentage of successful navigation trials as a function of trial duration. Higher is better.

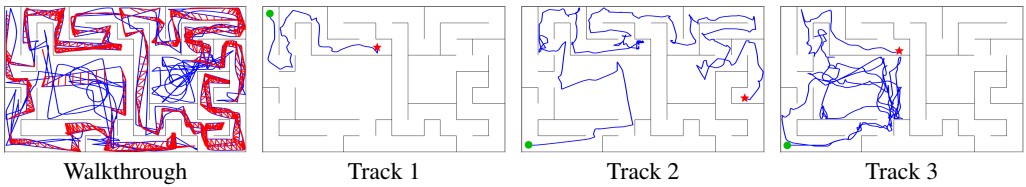

Figure 6: A walkthrough trajectory (left) and three goal-directed navigation tracks in the Val-3 maze (right). In the walkthrough trajectory, the shortcuts automatically found in the SPTM graph are shown in red. Goal-directed navigation trials shown in Tracks 1, 2, and 3 were all successful, but Track 3 was excessively long. Start positions are shown in green, goals in red.

Why is the performance of the baseline approaches in our experiments significantly weaker than reported previously (Mirowski et al., 2017)? The key reason is that we study generalization of agents to previously unseen environments, while Mirowski et al. (2017) train and evaluate agents in the same environment. The generalization scenario is much more challenging, but also more realistic. Our results indicate that existing methods struggle with generalization.

Interestingly, the best-performing baseline is goal-agnostic, not goal-directed. We see two main explanations for this. First, generalization performance has high variance and may be dominated by spurious correlations in the appearance of training and test mazes. Second, even in the training environments the goal-directed baselines do not necessarily outperform the goal-agnostic ones, since the large reward for reaching the goal makes reinforcement learning unstable. This effect has been observed by Mirowski et al. (2017), and to avoid it the authors had to resort to reward clipping; in our setting, reward clipping would effectively lead to ignoring the goals.

Figure 6 (left) shows a trajectory of a walkthrough provided to the algorithms in the Val-3 maze. The shortcut connections made automatically in the SPTM graph are marked in red. We selected a conservative threshold for making shortcut connections to ensure that there are no false positives. Still, the automatically discovered shortcut connections greatly increase the connectivity of the graph: for instance, in the Val-3 maze the average length of the shortest path to the goal, computed over all nodes in the graph, drops from 990 to 155 steps after introducing the shortcut connections.

Figure 6 (right) demonstrates three representative trajectories of the SPTM agent performing goal-directed navigation. In Tracks 1 and 2, the agent deliberately goes for the goal, making use of the environment representation stored in SPTM. Track 3 is less successful and the agent's trajectory contains unnecessary loops; we attribute this to the difficulty of vision-based self-localization in large environments.

Table 2 reports an ablation study of the SPTM agent on the validation set. Removing vision-based shortcuts from the graph leads to dramatic decline in performance. The agent with independent per-frame localization performs quite well on two of the three mazes, but underperforms on the

|                                | Val-1 | Val-2 | Val-3 |
|--------------------------------|-------|-------|-------|
| Ours – no visual shortcuts     | 85    | 55    | 50    |
| Ours – per-frame localization  | 95    | 87    | 52    |
| Ours – full                    | **100** | **98** | **100** |

Table 2: Ablation study on the SPTM agent. We report the percentage of navigation trials successfully completed in 5,000 steps in validation mazes (higher is better).

more challenging Val-3 maze. A likely explanation is that perceptual aliasing gets increasingly problematic in larger mazes.

Additional experiments are reported in the supplement: performance in the validation environments, robustness to hyperparameter settings, an additional ablation study evaluating the performance of the $R$ and $L$ networks compared to simple alternatives, experiments in environments with homogeneous textures, and experiments with automated (non-human) exploration.

## 5 CONCLUSION

We have proposed semi-parametric topological memory (SPTM), a memory architecture that consists of a non-parametric component – a topological graph, and a parametric component – a deep network capable of retrieving nodes from the graph given observations from the environment. We have shown that SPTM can act as a planning module in a navigation system. This navigation agent can efficiently reach goals in a previously unseen environment after being presented with only 5 minutes of footage. We see several avenues for future work. First, improving the performance of the networks $R$ and $L$ will directly improve the overall quality of the system. Second, while the current system explicitly avoids using ego-motion information, findings from experimental psychology suggest that noisy ego-motion estimation and path integration are useful for navigation. Incorporating these into our model can further improve robustness. Third, in our current system the size of the memory grows linearly with the duration of the exploration period. This may become problematic when navigating in very large environments, or in lifelong learning scenarios. A possible solution is adaptive subsampling, by only retaining the most informative or discriminative observations in memory. Finally, it would be interesting to integrate SPTM into a system that is trainable end-to-end.

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

## SUPPLEMENTARY MATERIAL

## S1 METHOD DETAILS

### S1.1 NETWORK ARCHITECTURES

The retrieval network $R$ and the locomotion network $L$ are both based on ResNet-18 (He et al., 2016). Both take $160 \times 120$ pixel images as inputs. The networks are initialized as proposed by He et al. (2016). We used an open ResNet implementation: `https://github.com/raghakot/keras-resnet/blob/master/resnet.py`.

The network $R$ admits two observations as input. Each of these is processed by a convolutional ResNet-18 encoder. Each of the encoders produces a 512-dimensional embedding vector. These are concatenated and fed through a fully-connected network with 4 hidden layers with 512 units each and ReLU nonlinearities.

The network $L$ also admits two observations, but in contrast with the network $R$ it processes them jointly, after concatenating them together. A convolutional ResNet-18 encoder is followed by a single fully-connected layer with 7 outputs and a softmax. The 7 outputs correspond to all available actions: do nothing, move forward, move backward, move left, move right, turn left, and turn right.

### S1.2 TRAINING

We implemented the training in Keras (Chollet et al., 2015) and Tensorflow (Abadi et al., 2016). The training setup is similar for both networks. We generate training data online by executing a random agent in the environment, and maintain a replay buffer $B$ of size $|B| = 10,000$. We run the random agent for 10,000 steps and then perform 50 mini-batch iterations of training. For the random agent, as well as for all other agents, we use action repeat of 4 – that is, every selected action is repeated 4 times. At each training iteration, we sample a mini-batch of 64 training observation pairs at random from the buffer, according to the conditions described in Sections 3.2 and 3.1. We then perform an update using the Adam optimizer (Kingma & Ba, 2015), with learning rate $\lambda = 0.0001$, momentum parameters $\beta_1 = 0.9$ and $\beta_2 = 0.999$, and the stabilizing parameter $\varepsilon = 10^{-8}$.

## S2 BASELINE DETAILS

The baselines are based on an open A3C implementation: `https://github.com/pathak22/noreward-rl`. We have used the architectures of Mnih et al. (2016) and Mirowski et al. (2017). The feedforward model consists of two convolutional layers and two fully-connected layers, from which the value and the policy are predicted. In the LSTM model the second fully connected layer is replaced by LSTM. The input to the networks is a stack of 4 most recent observed frames, resized to $84 \times 84$ pixels. We experimented with using RGB and grayscale frames, and found the baselines trained with grayscale images to perform better. We therefore always report the results for baselines with grayscale inputs. We train the baselines for 80 million action steps, which corresponds to 320 million simulation steps because of action repeat. We selected the snapshot to be used at test time based on the training reward.

## S3 ADDITIONAL RESULTS

Layouts of the validation mazes are shown in Figure S1. Plots of success rate as a function of trial duration on each validation maze are shown in Figure S2. Performance of an SPTM agent with varying hyperparameters is shown in Table S1.

### S3.1 HOMOGENEOUS TEXTURES AND AUTOMATED EXPLORATION

To evaluate the robustness of the approach, we tried varying the texture distribution in the environment and the properties of the exploration sequence.

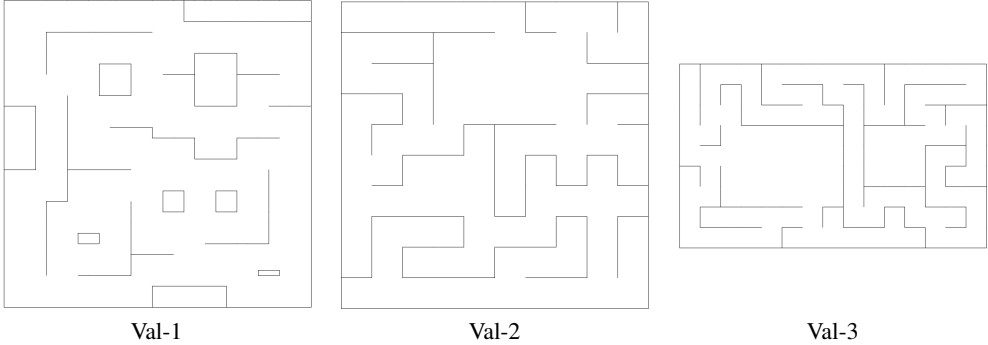

Figure S1: Layouts of the mazes used for validation.

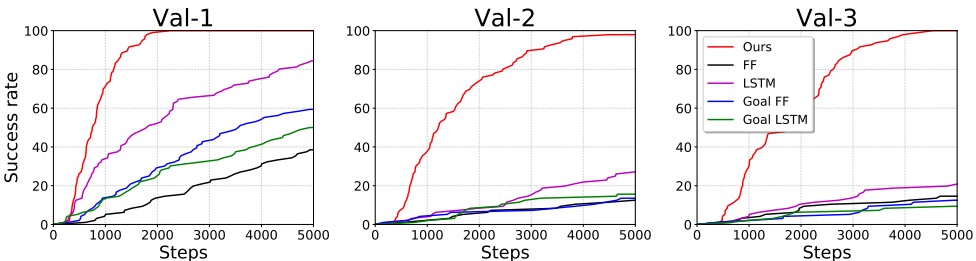

Figure S2: Percentage of successful navigation trials as a function of trial duration, in the validation mazes. Higher is better.

In the experiments in the main paper we used mazes with relatively diverse (although repetitive) textures, see for example Figure 3 in the main paper. We re-textured several mazes to be qualitatively similar to Mirowski et al. (2017): with mainly homogeneous textures and only relatively sparse inclusions of more discriminative textures. When testing the method with these textures, we re-trained the networks $R$ and $L$ in a training maze with similar texture distribution, but kept all other parameters of the method fixed.

For experiments in the main paper we used walkthrough sequences recorded from humans exploring the maze. An intelligent agent should be able to explore and map an environment fully autonomously. Effective exploration is a challenging task in itself, and a comprehensive study of this problem is outside the scope of the present paper. However, as a first step, we experiment with providing our method with walkthrough sequences generated fully autonomously – by our baseline agents trained with reinforcement learning. This is only possible in simple mazes, where these agents were able to reach all goals. We used the best-performing baseline for each maze and repeated exploration multiple times, until all goals were located.

The results are reported in Table S2. The use of automatically generated trajectories leads to only a minor decrease in the final performance, although qualitatively the trajectories of the SPTM agent become much noisier (not shown). The different texture distribution affects the results more, since visual self-localization becomes challenging with sparser textures. Yet the method still performs quite well and outperforms the baselines by a large margin.

### S3.2 ADDITIONAL ABLATION STUDY

To better understand the importance of the locomotion and retrieval networks, we performed two experiments. First, we substituted the retrieval network $R$ with simple per-pixel matching. Second, we substituted actions predicted by the locomotion network $L$ by actions from the exploration sequence (teach-and-repeat). Note that this second approach uses information unavailable to our method – actions performed during the walkthrough sequence. It thus cannot be considered a proper baseline. We further discuss the exact settings and the results.

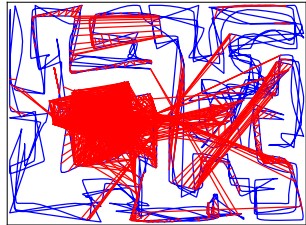 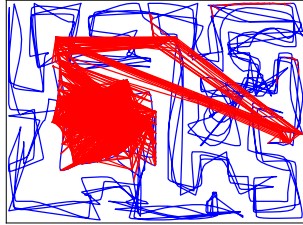

With normalization                    Without normalization

Figure S3: Graphs constructed using per pixel matching. Shortcut connections are shown in red. Most shortcuts connections are wrong – they connect distant locations.

### S3.2.1 PER-PIXEL COMPARISON

This experiment was inspired by the approach of Milford & Wyeth (2012). To compute the localization score, we downsample images to the resolution $40 \times 30$, convert to grayscale and then compute cosine distances between them. We experiment with two variants of this method: with local contrast normalization (similar to Milford & Wyeth (2012)) and without. To perform the normalization, we split the downsampled grayscale image into patches of size $10 \times 10$. In each patch, we subtract the mean and divide by the standard deviation.

As Table S3 indicates, the per-pixel comparison baseline performs poorly. As shown in Figure S3, the visual shortcuts made with this technique are catastrophically wrong. Local normalization only makes the results worse because it discards information about absolute color intensity, which can be a useful cue in our environments.

### S3.2.2 TEACH-AND-REPEAT

To be able to use actions from the exploration sequence, a few modification to our method are necessary. First, we introduce no shortcut connections in the graph, as we would not know the actions for them. The graph thus turns into a path, making the shortest paths longer. Second, to allow the agent to move along this path in both directions, we select an opposite for every action: for example, the opposite of moving forward is moving backward. Finally, we found that taking a fraction of completely random actions helps the agent not to get stuck when it diverges far from the exploration track and the recorded actions are not useful anymore. We found $10\%$ of random actions to lead to good results. Overall, the method works as follows. First, the goal and the agent are localized using the same procedure as our method. Then the agent has to move either forward or backward along the exploration graph-line. If forward, then the action corresponding to the agent's localized observation is taken, if backward – the opposite of the recorded action.

As Table S3 suggests, this method works significantly worse than our method, even though it makes use of extra information – the recorded actions. We see two reasons for this. First, there are no shortcut connections, which makes the path to the goal longer. Second, as soon as the agent diverges from the exploration trajectory, the actions do not match the states any more, and there is no mechanism for the agent to get back on track. For instance, imagine a long corridor: if the agent is oriented at a small angle to the direction of the corridor, it will inevitably crash into a wall. Why does the approach not fail completely due to the latter problem? This is most likely because the environment is forgiving: it allows the agent to slide along walls when facing them at an angle less than 90 degrees. This way, even if the agent diverges from the exploration path, it does not break down completely and still makes progress towards the goal. Videos of successful navigation trials for this agent can be found at `https://sites.google.com/view/SPTM`.

| Parameters | | | | | Environment | | |
|---|---|---|---|---|---|---|---|
| Loc. smooth. | #shortcuts | Mem. subsamp. | $s_{local}$ | $s_{reach}$ | Val-1 | Val-2 | Val-3 |
| 0 | 2000 | 4 | - | 0.95 | 95 | 88 | 52 |
| 10 | 2000 | 4 | 0.7 | 0.95 | 100 | 98 | 100 |
| 10 | 1000 | 4 | 0.7 | 0.95 | 99 | 92 | 90 |
| 10 | 2000 | 4 | 0.7 | 0.95 | 100 | 98 | 100 |
| 10 | 4000 | 4 | 0.7 | 0.95 | 100 | 98 | 100 |
| 1 | 2000 | 4 | 0.7 | 0.95 | 100 | 70 | 26 |
| 5 | 2000 | 4 | 0.7 | 0.95 | 100 | 96 | 99 |
| 10 | 2000 | 4 | 0.7 | 0.95 | 100 | 98 | 100 |
| 10 | 2000 | 4 | 0.7 | 0.9 | 100 | 95 | 100 |
| 10 | 2000 | 4 | 0.7 | 0.95 | 100 | 98 | 100 |
| 10 | 2000 | 4 | 0.7 | 0.97 | 100 | 97 | 100 |
| 10 | 2000 | 4 | 0.6 | 0.95 | 100 | 95 | 100 |
| 10 | 2000 | 4 | 0.7 | 0.95 | 100 | 98 | 100 |
| 10 | 2000 | 4 | 0.8 | 0.95 | 100 | 97 | 99 |
| 10 | 2000 | 1 | 0.7 | 0.95 | 76 | 47 | 73 |
| 10 | 2000 | 2 | 0.7 | 0.95 | 97 | 85 | 95 |
| 10 | 2000 | 4 | 0.7 | 0.95 | 100 | 98 | 100 |
| 10 | 2000 | 8 | 0.7 | 0.95 | 66 | 92 | 66 |
| 5 | 8000 | 1 | 0.7 | 0.95 | 99 | 84 | 93 |
| 10 | 4000 | 2 | 0.7 | 0.95 | 100 | 95 | 98 |
| 20 | 2000 | 4 | 0.7 | 0.95 | 100 | 98 | 100 |
| 40 | 1000 | 8 | 0.7 | 0.95 | 97 | 94 | 85 |

Table S1: Effect of hyperparameters, evaluated on the validation set. We report the percentage of navigation trials successfully completed in 5,000 steps (higher is better).

| | Val-3 | Test-1 | Test-4 | Test-5 |
|---|---|---|---|---|
| Ours - homogeneous textures | 55 | 98 | 76 | 75 |
| Ours - automated exploration | - | 94 | 93 | 91 |
| Ours - full | **100** | **100** | **100** | **100** |

Table S2: Evaluation of the SPTM navigation agent with homogeneous textures and automated exploration. We report the percentage of navigation trials successfully completed in 5,000 steps (higher is better).

| | Val-1 | Val-2 | Val-3 |
|---|---|---|---|
| Per-pixel comparison with normalization | 6 | 2 | 1 |
| Per-pixel comparison without normalization | 10 | 8 | 7 |
| Teach-and-repeat | 45 | 36 | 30 |
| Ours - full | **100** | **98** | **100** |

Table S3: Additional ablation study of the SPTM navigation agent. We report the percentage of navigation trials successfully completed in 5,000 steps in validation mazes (higher is better).

