# OpenReview forum: "Semi-parametric topological memory for navigation"
_ICLR.cc/2018/Conference — Accept (Poster)_

### Official Review · AnonReviewer1 · 2017-11-27
**Right approach -- poorly executed**

**Rating:** 3
**Confidence:** 4

**Review:**

The paper introduces a graph based memory for navigation agents. The memory graph is constructed using nearest neighbor heuristics based on temporal adjacency and visual similarity. The agent uses Dijkstra's algorithm to plan a a path through the graph in order to solve the navigation task.

There are several major problems with this paper. My overall impression is that the the proposed agent is a nearly hard-coded solution (which I think might be the correct approach to such problems), but a poorly implemented one. Specific points: 1-There are only 5 test mazes, and the proposed agent doesn't even solve all of them. 2-The way in which the maze is traversed in the exploration phase determines the accuracy of the graph that is constructed (i.e. traversing each location exactly once using a space-filling curve). 3-Of the two heuristics used in Equation 1 how many edges are actually constructed using the visual similarity heuristic? 4-How does the visual similarity heuristic handle visually similar map elements that correspond to distinct locations? 5- The success criteria of solving a maze is arbitrarily defined -- why exactly 2.4 min?

---

> ### Author Response · Authors · 2017-12-14
> **Authors' response**
>
> > My overall impression is that the the proposed agent is a nearly hard-coded solution (which I think might be the correct approach to such problems),
>
> The method uses more hand design than most end-to-end RL methods, but also includes crucial learned components. We believe that exploring the tradeoff between design and learning is important for solving complex problems such as mapping and navigation.
>
> > but a poorly implemented one.
>
> It would be helpful if the reviewer could be more specific here.
>
> > There are only 5 test mazes,
>
> This number of environments is similar to previous works on navigation, such as (Mirowski et al. 2017). Nevertheless, we will evaluate the methods in more environments.
>
> > the proposed agent doesn't even solve all of them.
>
> Typically a method does not have to “solve” all test cases it is applied to in order to be useful, but rather has to outperform relevant baselines. This is what we demonstrate. We are personally skeptical when a method works perfectly on all examples shown in a paper, as this typically indicates that either the problem was easy (not the case in our setting) or the examples are cherry-picked (likewise).
>
> > The way in which the maze is traversed in the exploration phase determines the accuracy of the graph that is constructed (i.e. traversing each location exactly once using a space-filling curve).
>
> Indeed the properties of the traversal affect the algorithm performance. This is, to our knowledge, the case for all mapping methods. We are not using space-filling curves, but rather simple human traversals.
>
> > Of the two heuristics used in Equation 1 how many edges are actually constructed using the visual similarity heuristic?
>
> Approximately 4000 shortcuts were made based on visual similarity. After introducing these shortcut connections, the average length of the shortest path to the goal, computed over all nodes in the graph, drops from 2500 to 772 steps. This indicates that the created shortcuts contribute significantly to the success of navigation. We will evaluate the method without these shortcuts and include the results in the paper.
>
> > How does the visual similarity heuristic handle visually similar map elements that correspond to distinct locations?
>
> This issue, known as perceptual aliasing, is a fundamental problem for all mapping and navigation algorithms. The basic version of our approach, presented in the paper, is indeed unable to disambiguate two identical observations. We implemented a simple modification of the approach: when localizing itself in the graph, the agent first searches in a neighborhood of the previous location, and only resorts for localizing itself in the whole graph if no confident match is found in this first step. This introduces temporal smoothness to localization of the agent, partially addresses the  perceptual aliasing problem, and improves the performance of the method, especially in complex mazes. We will add a comparison of the naive and the advanced versions of the method to the paper. More principled treatment of perceptual aliasing is an important direction of future work.
>
>  > The success criteria of solving a maze is arbitrarily defined -- why exactly 2.4 min?
>
> There has to be some upper limit on the duration of a navigation trial, and we chose 5000 simulation steps in this work. Plots in Figure 5 show the success rate as a function of episode duration for durations less than this maximum threshold.

---

### Official Review · AnonReviewer3 · 2017-11-27
**Exciting idea but disappointing implementation (revised)**

**Rating:** 7
**Confidence:** 5

**Review:**

*** Revision: based on the author's work, we have switched the score to accept (7) ***

Clever ideas but not end-to-end navigation.

This paper presents a hybrid architecture that mixes parametric (neural) and non-parametric (Dijkstra's path planning on a graph of image embeddings) elements and applies it to navigation in unseen 3D environments (Doom). The path planning in unseen environments is done in the following way: first a human operator traverses the entire environment by controlling the agent and collecting a long episode of 10k frames that are put into a chain graph. Then loop closures are automatically detected using image similarity in feature space, using a localization feed-forward ResNet (trained using a DrLIM-like triplet loss on time-similar images), resulting in a topological graph where edges correspond to similar viewpoints or similar time points. For a given target position image and agent start position image, a nearest neighbor search-powered Dijkstra path planning is done on the graph to create a list of waypoint images. The pairs of (current image, next waypoint) images are then fed to a feed-forward locomotion (policy) network, trained in a supervised manner.

The paper does not discuss at all the problems arising when the images are ambiguous: since the localisation network is feed-forward, surely there must be images that are ambiguously mapped to different graph areas and are closing loops erroneously? The problem is mitigated by the fact that a human operator controls the agent, making sure that the agent's viewpoint is clear, but the method will probably fail if the agent is learning to explore the maze autonomously, bumping into walls and facing walls. The screenshots on Figure 3 suggest that the walls have a large variety of textures and decorations, making each viewpoint potentially unique, unlike the environments in (Mirowski et al, 2017), (Jaderberg et al, 2017) and (Mnih et al, 2016).

Most importantly, the navigation is not based on RL at all, and ignores the problem of exploration altogether. A human operator labels 10k frames by playing the game and controlling the agent, to show it how the maze looks like and what are the paths to be taken. As a consequence, comparison to end-to-end RL navigation methods is unclear, and should be stressed upon in the manuscript. This is NOT a proper navigation agent.

Additional baselines should be evaluated: 1) a fully Dijkstra-based baseline where the direction of motion of the agent along the edge is retrieved and used to guide the agent (i.e., the policy becomes a lookup table on image pairs) and 2) the same but the localization network replaced by image similarities in pixel space or some image descriptor space (e.g., SURF, ORB, etc…). It seems to me that those baselines would be very strong.

Another baseline is missing: (Oh et al, 2016, “Control of Memory, Active Perception, and Action in Minecraft”).

The paper is not without merit: the idea of storing experiences in a graph and in using landmark similarity rather than metric embeddings is interesting. Unfortunately, that episodic memory is not learned (e.g., Neural Turing Machines or Memory Networks).

In summary, just like the early paper released in 2010 about Kinect-based RGBD SLAM: lots of excitement but potential disappointment when the method is applied on an actual mobile robot, navigating in normal environments with visual ambiguity and white walls. The paper should ultimately be accepted to this conference to provide a baseline for the community  (once the claims are revised), but I street that the claims of learning to navigate in unseen environments are unsubstantiated, as the method is neither end-to-end learned (as it relies on human input and heuristic path planning) nor capable of exploring unseen environments with visual ambiguity.

---

> ### Author Response · Authors · 2017-12-14
> **Authors' response - Part 1**
>
> We thank the reviewer for the thoughtful review and useful comments. We will update the paper accordingly.
>
> > feed-forward locomotion (policy) network, trained in a supervised manner.
>
> We would like to highlight that the locomotion policy is trained in self-supervised fashion, without any human labels or demonstration.
>
> > the problems arising when the images are ambiguous
>
> This issue, known as perceptual aliasing, is a fundamental problem for all mapping and navigation algorithms. The basic version of our approach, presented in the paper, is indeed unable to disambiguate two identical observations. We implemented a simple modification of the approach: when localizing itself in the graph, the agent first searches in a neighborhood of the previous location, and only resorts for localizing itself in the whole graph if no confident match is found in this first step. This introduces temporal smoothness to localization of the agent, partially addresses the  perceptual aliasing problem, and improves the performance of the method, especially in complex mazes. We will add a comparison of the naive and the advanced versions of the method to the paper. More principled treatment of perceptual aliasing is an important direction of future work.
>
> > The problem is mitigated by the fact that a human operator controls the agent, making sure that the agent's viewpoint is clear, but the method will probably fail if the agent is learning to explore the maze autonomously, bumping into walls and facing walls
>
> It is true that the method can be sensitive to the exploration sequence, and there is room for improvement. However, we would like to point out that 1) in this paper we focus on the memory architecture, 2) any mapping method would be sensitive to the properties of the walkthrough sequence, 3) availability of a short human exploration sequence is a reasonable assumption in many practical settings, for instance for a household robot, and 4) the baselines get access to exactly the same walkthrough sequence.
>
> To analyze the robustness of the method to the properties of the walkthrough sequence, we will perform experiments with non-human exploration and include the results in the paper.
>
> > The screenshots on Figure 3 suggest that the walls have a large variety of textures and decorations, making each viewpoint potentially unique, unlike the environments in (Mirowski et al, 2017), (Jaderberg et al, 2017) and (Mnih et al, 2016).
>
> It is true that our environments are different from the aforementioned DM Lab ones in terms of the texture distribution. This video https://youtu.be/4QxO8mdOf3M shows a walkthrough sequence of the Test-Difficult maze (other mazes are similar in texture distribution). Indeed the textures are diverse, but they are also repetitive, so the same seemingly discriminative texture appears in multiple locations in the maze. Moreover, the floor and the ceiling textures are uniform. This makes localization challenging. In the DM Lab environments, on the other hand, the floor textures vary, and walls are populated with unique “markers”.
>
> The textures we used were procedurally and randomly placed on the walls. We agree that additional evaluation on DM Lab environments or qualitatively similar ones in ViZDoom would be interesting and will work on adding these results to the paper.

---

> > ### Comment · AnonReviewer3 · 2017-12-26
> > **Waiting on additional results**
> >
> > Following the rebuttal, we are waiting on additional experiments with DMLab as well as non-human maze exploration of the maze (while the sequence of observations from human demonstration is not explicitly "labelled" information, it is nevertheless privileged information that an untrained RL agent or a random agent do not possess). I apologize if I have missed these additional results in the revision from 26 December.

---

> > > ### Author Response · Authors · 2017-12-27
> > > **Clarification**
> > >
> > > Thanks for your response! For us to address the reviewer’s concerns properly, it would be helpful if the reviewer could clarify a bit.
> > >
> > > We are still unsure: what exactly is the unfairness of using human demonstration the reviewer is mentioning? At training time none of the methods has access to human demonstrations. At test time all methods do: for the baselines, we feed the exploration sequence before starting the navigation trial, and the LSTM can potentially remember the maze layout. What exactly is unfair about this setup? Which experiment would get rid of this unfairness?
> > >
> > > Note that reliable autonomous exploration of a previously unseen environment is, to our knowledge, an unsolved problem. A random agent, an untrained RL agent, or even an RL agent trained in other environments, would not fit, since they will typically fail to explore the whole maze. Our contribution is in memory, not exploration. We compare different types of memory by providing them with the same (human) walkthrough exploration sequence. We believe that proper exploration requires memory, and now SPTM can be used to start attacking this problem.
> > >
> > > We are working on experiments in DMLab-like environments.

---

> > > ### Author Response · Authors · 2018-01-05
> > > **Additional results**
> > >
> > > We have performed the requested experiments and described them in the rebuttal, please see at the top of the comment section. In brief, both with ambiguous DMLab-like textures and with non-human exploration the method still works well, although somewhat worse compared to visually rich environments with human exploration.

---

> > > > ### Comment · AnonReviewer3 · 2018-01-11
> > > > **Thank you for the additional experiments**
> > > >
> > > > Taking into account the revision, this is an interesting idea whose limitations have been properly investigated.

---

> ### Author Response · Authors · 2017-12-14
> **Authors' response - Part 2**
>
> > Most importantly, the navigation is not based on RL at all, and ignores the problem of exploration altogether.
>
> In this work we focus on memory, not exploration. In fact, exploration of a previously unseen environment is a very difficult problem of its own, which itself needs memory: in order to know which parts of the environment to explore, the agent needs to know where it is currently located and which parts had already been explored. In this work, we make a useful contribution by designing the memory module. Using it for exploration (potentially in combination with RL) is a very exciting avenue for future work.
>
> The proposed method can be seen as an advanced version of visual imitation learning (the agent does not have access to the expert’s actions!) combined with mapping. Interestingly, a concurrent submission to ICLR proposes to use a method very similar to our locomotion network for visual imitation learning: https://openreview.net/forum?id=BkisuzWRW .
>
> > A human operator labels 10k frames by playing the game and controlling the agent, to show it how the maze looks like and what are the paths to be taken. As a consequence, comparison to end-to-end RL navigation methods is unclear, and should be stressed upon in the manuscript.
>
> There seems to be some misunderstanding here. The operator does not “label” the images: the actions taken by the human are not available to the agent, only the observed image sequence. This image sequence is provided both to our agent and the baseline agents at test time. At training time, none of the agents have access to human demonstrations. Therefore, the comparison is fair.
>
> > a fully Dijkstra-based baseline where the direction of motion of the agent along the edge is retrieved and used to guide the agent
>
> If we understand the suggestion correctly, it requires recording the expert’s actions during the walkthrough and then repeating them at test time. As noted above, in our setup the actions from the walkthrough sequence *are not available* to the agents. Therefore this baseline has access to more information than the methods evaluated in the paper. Nevertheless, we will work on evaluating this baseline and including it in the paper.
>
> > the localization network replaced by image similarities in pixel space or some image descriptor space (e.g., SURF, ORB, etc…).
>
> Thanks for this suggestion! These are indeed very meaningful baselines. We will evaluate and add the results to the paper.
>
> > Oh et al, 2016, “Control of Memory, Active Perception, and Action in Minecraft”).
>
> Thanks for pointing this out, the work of Oh et al. is definitely very relevant. We will add a discussion to the paper. However, note that experiments by Oh et al. are performed in gridworld-like environments, in contrast with large mazes with continuous state space in our work. We expect that scaling the method of Oh et al. to these environments will be challenging. We will compare to the work of Oh et al. and will include the results in the paper.
>
> > Unfortunately, that episodic memory is not learned (e.g., Neural Turing Machines or Memory Networks).
>
> Indeed the memory is not learned end-to-end via RL, but both the embedding and the locomotion network are learned in self-supervised fashion. We strongly believe that both approaches have value, since RL alone provides only a relatively weak training signal. Combining the two is an exciting avenue for future work.
>
> > potential disappointment when the method is applied on an actual mobile robot, navigating in normal environments with visual ambiguity and white walls.
>
> We agree that deployment on a real robot would be challenging, which almost certainly would also be the case for any other navigation algorithm developed in a simulated environment. However, we are happy to see that the independent concurrent approach mentioned earlier (https://openreview.net/forum?id=BkisuzWRW), which is similar to our locomotion network, can be deployed on a real robot.
>
> > but I street that the claims of learning to navigate in unseen environments are unsubstantiated, as the method is neither end-to-end learned (as it relies on human input and heuristic path planning) nor capable of exploring unseen environments with visual ambiguity.
>
> We never aimed to state that the proposed method learns navigation and exploration end-to-end. In fact, the availability of a walkthrough video, as well as our focus on the memory module, are mentioned both in the abstract and in the introduction. We will carefully review the paper to make sure no inappropriate claims are made. If there are any specific parts the reviewer would like us to revise, it would be very helpful to point those out.

---

### Official Review · AnonReviewer2 · 2017-11-29
**Interesting approach with compelling results but is contrasted against weaker baselines**

**Rating:** 7
**Confidence:** 4

**Review:**

The paper presents for learning to visually navigate using a topological map is presented. The method combines an algorithmic approach to generate topological graphs, which is indexed by observations with Dijkstra's algorithm to determine a global path from which a waypoint observation is selected. The waypoint along with the current observation is fed to a learned local planner that transitions the agent to the target waypoint. A learned observation similarity mapping localizes the agent and indexes targets in the graph.

The novelty of the approach in context of prior visual navigation and landmark-based robotics research is the hybrid algorithmic and learned approach that builds a graph purely from observations without ego motion or direct state estimation. Most of the individual components have previously appeared in the literature including graph search (also roadmap methods), learned similarity metrics, and learned observation-based local planners. However, the combination of these approaches along with some of the presented nuanced enhancements including the connectivity shortcuts (a simple form of loop closure) to visual topological navigation and are compelling contributions. The results while not thorough do support the ability of the method effectively plan on novel and unseen environments.

The approach has potential limitations including the linear growth in the size of the memory, and it is unclear how the method handles degenerate observations (e.g., similar looking hallways on opposite sides of the environment). The authors should consider including examples or analysis illustrating the method’s performance in such scenarios.

The impact of the proposed approach would be better supported if compared against stronger baselines including recent research that address issues with learning long sequences in RNNs. Furthermore, additional experiments over a greater number and more diverse set of environments along with additional results showing the performance of the method based on variation environment parameters including number of exploration steps and heuristic values would help the reader understand the sensitivity and stability of the method.

The work in “Control of Memory, Active Perception, and Action in Minecraft” by Oh et al. have a learned memory that is used recall previous observations for planning. This method’s memory architecture can be considered to be nonparametric, and, while different from the proposed method, this similarity merits additional discussion and consideration for empirical comparison.

Some of the the details for the baseline methods are unclear. The reviewer assumes that when the authors state they use the same architecture as Mnih et al. and Mirowski et al. that this also includes all hyper parameters including the size of each layer and training method. However, Mirowski et al. use RGB while the stated baseline is grayscale.

The reviewer wonders whether the baselines may be disadvantaged compared to the proposed method. The input for the baselines are restricted to a 84x84 input image in addition to being grayscale vs the proposed methods 160x120 RGB image. It appears the proposed method is endowed with a much greater capacity with a ResNet-18 in the retrieval network compared to the visual feature layers (two layers of CNNs) of the baseline networks. Finally the proposed method is provided with demonstrations (the human exploration) that effectively explore the environment. The authors should consider bootstrapping the baseline methods with similar experience.

---

> ### Author Response · Authors · 2017-12-14
> **Authors' response**
>
> We thank the reviewer for the thoughtful review and the useful comments. We will update the paper accordingly.
>
> > the proposed method is provided with demonstrations (the human exploration) that effectively explore the environment. The authors should consider bootstrapping the baseline methods with similar experience.
>
> We believe there might be a misunderstanding of our experimental setup here. When testing the baselines, we feed them the walkthrough sequence (no expert actions, only the observations!) before each navigation trial. LSTM could remember the environment layout based on this sequence. Both the proposed method and the baselines have access to the walkthrough sequence at test time, and none of the methods have access to them at training time. Therefore, the comparison is fair.
>
> > the linear growth in the size of the memory
>
> This is indeed an undesirable property, and future work will have to address this. As a first step, we will include in the paper experiments with memory sub-sampled by a constant factor.
>
> > handling degenerate observations (e.g., similar looking hallways on opposite sides of the environment)
>
> This issue, known as perceptual aliasing, is a fundamental problem for all mapping and navigation algorithms. The basic version of our approach, presented in the paper, is indeed unable to disambiguate two identical observations. We implemented a simple modification of the approach: when localizing itself in the graph, the agent first searches in a neighborhood of the previous location, and only resorts for localizing itself in the whole graph if no confident match is found in this first step. This introduces temporal smoothness to localization of the agent, partially addresses the perceptual aliasing problem, and improves the performance of the method, especially in complex mazes. We will add a comparison of the naive and the advanced versions of the method to the paper. More principled treatment of perceptual aliasing is an important direction for future work.
>
> > Discuss and evaluate “Control of Memory, Active Perception, and Action in Minecraft” by Oh et al.
>
> Thanks for pointing this out, the work of Oh et al. is definitely very relevant. We will add a discussion to the paper. However, note that experiments by Oh et al. are performed in gridworld-like environments, in contrast with large mazes with continuous state space in our work. We expect that scaling the method of Oh et al. to these environments will be challenging. We will compare to the work of Oh et al. and will include the results in the paper.
>
> > compare against stronger baselines including recent research that address issues with learning long sequences in RNNs
>
> We would gladly compare to more elaborate memory architectures, if the reviewer could point at existing implementations that use these memory architectures for navigation in environments with continuous state spaces. Implementing and tuning such a navigation system from scratch would be a complex undertaking of its own.
>
> > Furthermore, additional experiments over a greater number and more diverse set of environments
>
> We will evaluate the methods in additional environments and will include the results in the paper.
>
> > results showing the performance of the method based on variation environment parameters and heuristic values
>
> We agree that such an analysis would be very useful. We will run corresponding experiments and include the results in the paper.
>
> > Hyperparameters of baselines
>
> For the baseline evaluation, we kept the hyperparameters as similar as possible to Mirowski et al.
>
> > Grayscale for the baselines, RGB for the proposed method
>
> Indeed, we fed grayscale images to the baselines instead of RGB. According to our previous experience, RL navigation methods typically do not benefit from RGB images. Still, we agree that for fair evaluation RGB images should be used, and we will re-evaluate the baselines with RGB images.
>
> > 84x84 input image and small network for baselines, 160x120 image and ResNet for the proposed method
>
> The proposed method does make use of a higher resolution image and a larger network. However, we are not aware of any existing works demonstrating training of ResNet-size networks with large input images from scratch using RL. Moreover, such experiments could get extremely slow. 84x84 images and small networks are the absolute standard in the deep RL literature, and therefore we assumed it is fair to use these standard settings for the baselines. We will add baseline experiments with 160x120 input images.

---

### Author Response · Authors · 2017-12-14
**Response to the reviewers**

We thank the reviewers for their work and their valuable comments. We are happy that the reviewers find the idea of topological memory exciting and compelling (AR2, AR3), note that this might be the right approach to the problem (AR1), and argue that the paper should be ultimately accepted for publication after revising the text (AR2, AR3).

We respond in detail to each of the reviewers individually in comments to their reviews. We will work on performing the proposed experiments and updating the paper accordingly.

We realized that the paper may not give a good impression of the appearance of the mazes and the difficulty of the task. This link https://youtu.be/4QxO8mdOf3M shows the human walkthrough sequence for the Test-Difficult maze. Note that the textures are diverse, but repetitive. Other mazes have similar texture distribution.

---

### Author Response · Authors · 2017-12-26
**Paper revision**

We have submitted a revision of the paper. Key changes:
- Added experiments in 5 more environments. There are now 3 validation and 7 test mazes. The proposed agent outperforms the baselines by a large margin in all of them.
- Added an improved localization technique with temporal smoothing, partially addressing the perceptual aliasing problem.
- Added an analysis of hyperparameter importance. The method is generally robust to hyperparameter values.
- Improved the evaluation (16 trials per maze -> 96 trials per maze).
- Demonstrated that SPTM agent performs well when the memory is temporally downsampled by a factor up to 4.

We continue working on other experiments requested by the reviewers, and will add them to the later revisions of the paper.

---

### Author Response · Authors · 2018-01-05
**Authors’ rebuttal**

To address the reviewers’ concerns, we performed multiple additional experiments. First, as described in our previous comment, we ran an extensive evaluation in five more mazes and with variable hyperparameters, clearly demonstrating the advantage of our method over baselines. We also introduced an improved localization method which can better deal with perceptual aliasing. All these results are already shown and discussed in the revised paper.

More recently, we performed two more series of experiments requested by the reviewers: on navigation in DMLab-like environments with ambiguous textures, and on non-human exploration. These experiments show that our method can work even with sparse landmarks and non-human exploration. These results are not yet discussed in the paper, but we will add them to the final version. Note that operation in the ambiguous textures scenario is made possible by the simple improvement to agent’s localization: instead of localizing independently per frame, we smooth the self-localization of the agent by biasing it towards localization from the previous step (see Sec. 3.1, “Finding the waypoint”, for a description). We now describe these experiments in more detail.

In the first series of experiments, we re-created in Vizdoom the environments from DMLab used by Mirowski et al. with exactly the same layout and similar sparsity of landmarks. We additionally used 2 mazes from our own set with DMLab-like texture distribution, bringing the number of mazes to a total of 4. The walkthroughs for those environments are shown in the following videos: https://youtu.be/MS8ReBruns4 , https://youtu.be/Jlhkc-9hZdo , https://youtu.be/Ko5-SB6CpyQ , https://youtu.be/Rlgl01S7PQQ . Those environments are highly ambiguous: walls are mostly covered with a single texture, with occasional distinct textures.
Results: we report success rate @5000 steps, higher is better. In the mazes DMLab small (Test-1 in our notation), DMLab large (Val-3 in our notation), Test-5, Test-4 the method achieves the following results (respectively):
89%, 44%, 68%, 48%.
This can be compared to the results of our method in the corresponding visually rich environments, as reported in the paper:
100%, 97%, 76%, 89%.
The results of best RL-based baselines are (respectively):
53%, 21%, 32%, 25%.
Clearly, in ambiguous environments the performance of the proposed method drops, but the method does not break down completely and still outperforms the baselines by a large margin. Videos of navigating agent are available at https://youtu.be/NpfAF6LILXc , https://youtu.be/cZ5RMGs4LE8 , https://youtu.be/8ehxeeHiK-E . We point out that it is incorrect to assume that ambiguous environments are more realistic than visually rich environments from our previous experiments: both types of environments exist in real world. Yet, we admit that these visually ambiguous environments remain challenging for our method, and in future work we plan to further improve the model so as to better deal with these.

In the second experiment, we aimed to verify that our method still works even when given inefficient non-human exploration. For that purpose, we used traversals from the RL-based baseline with LSTM memory (see the paper). Of course, this only works for simple environments. Even for these, most traversals don't go through all the goals. To avoid this problem, we sampled the traversals until they covered all the goals. It took roughly 20 attempts for the mazes we tried. Here we report the results on 2 mazes: Test-1, Test-4. Those were chosen as the simplest so that the exploration heuristic works. The results are:
63%, 52%
For human traversal before:
100%, 89%
The best baselines were:
53%, 25%
As can be seen, our method does not break here either, still beats the baselines, although by a smaller margin. Obviously, the agent's tracks get longer as the inefficient exploration wastes time instead of moving between different locations quickly (for example, see exploration tracks here: https://youtu.be/NHjqQS-d_ko , https://youtu.be/cEaaPVUu17I ).

We hope these results persuade the reviewers that the proposed method is generally applicable and promising. We will continue working on the remaining experiments proposed by the reviewers.

---

### Decision · Program_Chairs · 2018-01-29
**ICLR 2018 Conference Acceptance Decision**

**Decision:**

Accept (Poster)

**Comment:**

Important problem (navigation in unseen 3D environments, Doom in this case), interesting hybrid approach (mixing neural networks and path-planning). Initially, there were concerns about evaluation (proper baselines, ambiguous environments, etc). The authors have responded with updated experiments that are convincing to the reviewers. R1 did not participate in the discussion and their review has been ignored. I am supportive of this paper.